# Intestinal Explant Cultures from Gilthead Seabream (*Sparus aurata, L.)* Allowed the Determination of Mucosal Sensitivity to Bacterial Pathogens and the Impact of a Plant Protein Diet

**DOI:** 10.3390/ijms21207584

**Published:** 2020-10-14

**Authors:** David Sánchez Peñaranda, Christine Bäuerl, Ana Tomás-Vidal, Miguel Jover-Cerdá, Guillem Estruch, Gaspar Pérez Martínez, Silvia Martínez Llorens

**Affiliations:** 1Aquaculture and Biodiversity Research Group, Institute of Science and Animal Technology (ICTA), Universitat Politècnica de València, 46022 Valencia, Spain; atomasv@dca.upv.es (A.T.-V.); mjover@dca.upv.es (M.J.-C.); guiescu@etsia.upv.es (G.E.); silmarll@dca.upv.es (S.M.L.); 2Department of Biotechnology, Institute of Agrochemistry and Food Technology, Consejo Superior de Investigaciones Científicas (CSIC) (Spanish National Research Council), 46980 Paterna, Valencia, Spain; cbauerl@iata.csic.es (C.B.); gaspar.perez@iata.csic.es (G.P.M.)

**Keywords:** gilthead seabream, ex vivo, intestine explants culture, RT-qPCR, inflammation, plant protein

## Abstract

The interaction between diet and intestinal health has been widely discussed, although in vivo approaches have reported limitations. The intestine explant culture system developed provides an advantage since it reduces the number of experimental fish and increases the time of incubation compared to similar methods, becoming a valuable tool in the study of the interactions between pathogenic bacteria, rearing conditions, or dietary components and fish gut immune response. The objective of this study was to determine the influence of the total substitution of fish meal by plants on the immune intestinal status of seabream using an ex vivo bacterial challenge. For this aim, two growth stages of fish were assayed (12 g): phase I (90 days), up to 68 g, and phase II (305 days), up to 250 g. Additionally, in phase II, the effects of long term and short term exposure (15 days) to a plant protein (PP) diet were determined. PP diet altered the mucosal immune homeostasis, the younger fish being more sensitive, and the intestine from fish fed short-term plant diets showed a higher immune response than with long-term feeding. *Vibrio alginolyticus* (*V. alginolyticus*) triggered the highest immune and inflammatory response, while COX-2 expression was significantly induced by *Photobacterium damselae* subsp. *Piscicida* (*P. damselae subsp. Piscicida*), showing a positive high correlation between the pro-inflammatory genes encoding interleukin 1*β* (*IL1-β*), interleukin 6 (*IL-6*) and cyclooxygenase 2(*COX-2*).

## 1. Introduction

In addition to the digestion and absorption of nutrients, the fish intestine is a complex biological system that represents a major defense barrier against pathogens and plays a crucial role in osmoregulation and immune and inflammatory response [1]. Furthermore, the intestine participates in the modulation of gastrointestinal microbiota, inducing inflammatory responses against pathogenic bacteria or developing immunotolerance to luminal bacteria.

The interactions between fish intestinal immunity, pathogenic bacteria and commensal microbiota in the gut have been widely reviewed [2]. Bacterial challenges in vivo require specialized settings, expensive operating costs, and a high number of fish, and it is difficult to perform them and achieve the desired experimental working conditions [3]. In this regard, systems have been developed, based on the ex vivo maintenance of intestine fragments, to evaluate successfully the effect of different bacterial strains on intestinal health and provide very reliable information on the interactions between the bacteria and the host. The ex vivo intestinal sack method [4] has been used to assess the histological and microbial changes in fish in response to bacteria exposure [3,5,6,7,8,9]; however, this method is highly restricted by the tissue’s viability under experimental conditions [3,4]. The development of new experimental models based on tissue explants culture has allowed the maintenance of tissue lifespan, as well as immune and histological features [10,11,12]. These systems have been used to register responses to exposure to specific bacteria in human tissue explant cultures, also at the gene expression level [12].

On the other hand, the necessity to replace fish meal with alternative protein sources in aquafeeds has led researchers to focus on the impact of the inclusion of alternative ingredients, such as plant protein sources. Although its use at high levels without impairing growth performance is feasible [13,14], negative effects on immune capacity, with higher mortalities, have been observed [15]. Previous findings revealed that during early growth stages, the inclusion of plant protein has a great impact [16,17]; at the intestinal level, the inclusion of plant sources in diets has been related to morphological alterations, changes in the intestinal bacterial community, inflammatory events, and a lack of capacity to regulate intestinal epithelial integrity [18]. Similar results have been reported in vivo at the molecular level, with an altered gene expression pattern [19,20,21].

Research related to the impact of fish meal replacement becomes even more relevant for carnivorous species, such as gilthead seabream. In this species, previous studies with different levels of substitution and alternative ingredients have been assayed to evaluate zootechnical parameters and survival [16,22,23], and the immune in vivo response to fish meal replacement [24,25,26] or bacterial infection [27,28,29,30].

The ex vivo response of intestinal tissue from fish feed with different protein sources to bacterial challenge has been previously addressed in other species [3]. However, to the best of our knowledge, this is the first study involving gene expression determination in fish intestinal explants after ex vivo bacterial exposure.

Therefore, in the current work, an intestine explant culture system was developed to evaluate a possible differential inflammatory and immune response to ex vivo bacterial challenge in fish after long-term total fish meal replacement at different stages of growth. Additionally, the effect of a short-term replacement of total fish meal with a plant mixture on growing fish was also assayed.

## 2. Results

With the aim to evaluate the effect of a plant protein (PP) diet on inflammatory and immune gene expression in the gut, 240 fish were fed on fish meal (FM) diet and PP for 305 days. The effect of the diet was investigated in a first stage at day 90 (phase I) and the experiment continued until day 305 (phase II). Two weeks before the end of the experiment, a group of animals from the FM group was introduced to the PP diet, in order to assess a possible short-term effect of the PP in adult fish. Intestine fragments of fish harvested in phase I and phase II were used for the ex vivo pathogen challenge assay, but also basal gene expression was determined on intestinal samples.

Optimal conditions for intestinal explant culture were set up through different approaches. Tissue and cellular integrity were monitored by the release of lactate dehydrogenase (LDH) activity at 0, 4, 6, and 24 h of incubation. No relevant differences were observed at the tissue level, but a significant increase in LDH was registered at 24 h of incubation in the explant culture medium (Appendix A). Candidate housekeeping genes for real time qPCR were tested at different times, indicating that 6 h was an appropriate time of incubation (Appendix A). On the other hand, if the effect of the ex vivo procedure is evaluated, significant differences were observed in most of the genetic markers analyzed (Appendix A). Therefore, in the following assays, gene expression was normalized based on the ex vivo unchallenged samples.

Finally, the consistency of the ex vivo assay for biological replicates was checked by a correlation analysis for each biological replicate pair (Appendix A). The adjustment to the lineal model was particularly good for pro-inflammatory genes (*Interleukin 1β: IL-1β; Interleukin 6:I L-6; Cyclooxygenase 2: COX-2).*

### 2.1. Phase I: Up to 68 g

With the aim of assessing the effect of a total fish meal substitution on intestinal inflammatory and immune status at early growth stages (68 g fish weight), basal gene expression was determined and ex vivo trials were carried out with 6 h of challenge of pathogenic bacteria cultures.

#### 2.1.1. Basal Gene Expression

After 90 days fed with PP and FM diets, fish intestines showed a significantly different gene expression profile, as the PP group had a higher expression level for *IL-1β* and lower for *COX-2* (Figure 1).

#### 2.1.2. Ex Vivo Assay

The explant culture system was used to determine the immune response of intestinal fragments from on-growing seabream specimens fed with different diets (Figure 2A). Additionally, the intestine was divided into two segments—foregut (FG) and hindgut (HG)—to evaluate a possible differential immune response (Figure 2B). At 6 h, gene expression was statistically different between the PP and FM diets in most of the markers, and the induction of *IL-1β* and in fish under the PP diet was particularly remarkable (Figure 2A). Gene expression responses of the different intestinal sections were quite similar, with a higher expression of *IL-6* in FG at 4 h (Appendix A) and Occludin (*Ocl*) at 6 h in HG (Figure 2B).

If only the bacteria variable is considered, significant differences were only observed with *V. alginolyticus* after 6 h of incubation, specifically with a higher expression of *COX-2* and *Immunoglobulin M (IgM)*, and notably of IL-1β (Figure 3A), indicating that this bacterium is able to induce the greatest response in the majority of the cases (Figure 3B–F). *P. damselae* subsp. *piscicida* reported the second highest response, being even a little greater than *V. alginolyticus* in some cases, such as for *IL-6* or *Ocl* genes. Taking into consideration the diet, fish belonging to the PP group showed, in general, a greater response to the bacterial stimuli (Figure 3), with significant differences for *IL-6*, *COX-2*, and *Ocl* genes. In general, this tendency to a higher response in the PP group was not observed at 4 h of incubation, registering significant differences only in the *Ocl* gene (Appendix A).

Finally, if a multifactorial analysis of variance is performed taking into account all the factors, *Ocl* and *IL-6* were significantly altered by the diet and the section at 4 h of incubation, respectively. Nevertheless, 6 h of exposition was necessary to register differences caused by bacterial stimulus (Appendix A).

### 2.2. Phase II: Up to 250 g

This second assay sought to evaluate the impact of a total substitution of fish meal at longer term feeding, up to 252 g (305 day; PP) on intestinal gene expression. Additionally, the differential response of total fish meal substitution with plant protein during a short-term period (15 days; PP*) was evaluated.

#### 2.2.1. Basal Gene Expression

No gene expression differences were observed between fish fed the FM diet and plant-based diets in the long term (PP) as well as in the short term (PP*), except for *IgM* in the PP group (Figure 4).

#### 2.2.2. Ex vivo Assay

The expression of pro-inflammatory genes (*IL-1*, *IL-6*, and *COX-2* genes) increased after 6 h of incubation in the ex vivo unchallenged group with respect to the basal values, confirming the results of the previous assay (Appendix A). Hence, expression results in samples incubated with the different bacteria were normalized with the expression of the control samples, for each experimental factor.

A multifactorial ANOVA of gene expression, taking into consideration the diet, section, and bacterial challenge (stimuli), underlined a significant linkage of the PP diet with *COX-2* and *Ocl*. As expected, the expression of pro-inflammatory genes (*IL-1β*, *IL-6*, and *COX-2*) was bound to the pathogen stimuli (Table 1), while again no differences were reported between sections. Therefore, due to the lack of statistical differences (*p* < 0.05) between intestinal sections, the following analyses were performed joining both sections.

The exposition to *V. alginolyticus* and *P. damselae* subsp. *piscicida* induced a remarkable increase in *IL-1β* with respect to unchallenged samples and, in fact, was able to induce significant *IL-1β* response in all diet groups (Figure 5B). Additionally, the exposition to the cited bacteria induced higher *IL-6* expression in the control group (FM) (Figure 5). Regarding the sensitivity to the pathogen as function of the diet, the PP* group showed a remarkable tendency to have higher expression values for all tested genes in response to bacteria, but only *COX-2* and *Ocl* showed significant differences with respect to FM and PP groups (Figure 5D,F).

Finally, as expected, there was a high correlation (Pearson’s coefficient) between the expression of IL-1β, a known master regulator of innate immune response and inflammation, and *IL-6* and *COX-2 (IL-1β/IL-6* = 0.74; *IL-1β/COX-2* = 0.72) (Figure 6).

## 3. Discussion

It has been extensively proven that the inclusion of alternative plant proteins can lead to nutritional imbalances [23,31] and immune dysfunctions [24,32], particularly in carnivorous fish, increasing their susceptibility to pathogenic invasion, disease, and finally, death. The purpose of this work was to confirm, in ex vivo conditions, the effect caused by total FM substitution for a long and short period on immune intestinal status. The findings of the present study suggest that the fish gut response to the total dietary substitution of fish meal by plant protein meals differs between short- and long-term feeding and the fish size. In addition, previous studies in vivo carried out with seabream demonstrated that the use of plant proteins induced significant alterations of the gut microbiota, gut gene expression, and gut proteomic profile [33,34,35]. Nonetheless, these alterations can be affected to a different extent by the feeding period or fish stage. In this study, the effect of total fish meal substitution by plant protein was evaluated in two stages of fish growth: from 12 to 68 g (phase I) and up to 252 g (phase II). In phase II, the effect on intestinal health status of FM substitution by PP throughout the whole period (305 days) was compared with that of a shorter term of feeding (15 days).

An explant culture assay has been implemented in this work to evaluate the intestinal health status of fish exposed to a PP diet. Ex vivo approaches based on explants culture proved to be useful to analyze pro-inflammatory responses [12]. In fish, several works have been attempted to evaluate the host-pathogen [36], especially by way of the intestinal sack method [3,7,8,9]. In the present work, significant differences were clearly observed with incubations of the intestine explants of 6 h, and these experimental conditions were shown to preserve tissue integrity and to trigger a detectable immune response. Furthermore, explants obtained from the same section of each fish demonstrated a sound consistency in the response to pathogens. Although a different immunological performance has been attributed to the FG and HG [37], using this experimental set up no significant differences could be found between gene expression in both segments. This allowed an increase in the number of explant fragments from each single intestine, thus improving the experimental efficiency and reducing the number of fish and, hence, reducing individual variability.

In the present experiment, three pro-inflammatory markers have been monitored to assess the inflammatory status (IL-1β, IL-6, COX-2) [38,39]. IgM is considered the most abundant immunoglobulin in plasma, high levels in fish fed with plant sources-based diets have been reported [40], and its expression was induced in mucosal tissues as a response to pathogen infection [41]. Finally, Ocl is a key protein in the regulation of tight junctions between enterocytes, and therefore, in the permeability of the epithelial barrier [42].

The stage of fish growth and the feeding period have a clear influence on the response of fish performance to dietary plant protein and previous studies reported that juvenile fish tolerate less dietary plant protein than commercial size fish [16,43]. In phase I, intestine explants of seabream fed with plant protein for 90 days showed a high basal inflammatory response, with higher expression of *IL-1β, IL-6,* and also *COX-2*. IL-1β is a known canonical master regulator of pro-inflammatory processes, it is secreted in response to Gram-negative bacteria, and its release is followed by the production of IL-6 and other cytokines in the pro-inflammatory cascade [38,44]. Modification in diet composition changed the expression of IL-1β, in agreement with previous studies, thus indicating that a PP rendered an already sensitized intestine susceptible to inflammatory/stress stimuli [35,44,45,46].

During infection, IL-1β is induced, and an increased expression of IL-1β has been reported in the intestine of different species, including gilthead seabream [29], after intraperitoneal challenge with gram negative bacteria. Increased expression of IL-1β and COX-2 has been reported before after in vitro challenge of gilthead seabream immune cells with bacteria or commercial pathogen-associated molecular pattern (PAMP) solutions [47,48,49,50]. Although COX-2 is a typical oxidative stress marker, its expression is also induced by inflammatory mediators [49,51], including IL-1β [52], as it takes part in the production of reactive oxygen species (ROS) and NO that have antibacterial activity and are also part of the innate immune system in higher vertebrates and carp macrophages [53,54]. In addition, genes related to the maintenance of epithelial tissue integrity, such as *Ocl* (expressing occludin), are influenced by inflammatory processes [55,56], and its regulation depends on several cytoskeletal, scaffolding, signaling, and polarity proteins [42], and it is definitely related to epithelial barrier functions in vivo and in vitro [56]. In mice, all these genes are possibly connected for the maintenance of the intestinal barrier [57].

In phase II (up to 250 g), differences were only found in the *IgM* gene between FM and PP experimental groups, supporting the hypothesis that older fish are more tolerant to diet plant proteins than fish in earlier growth stages [16]. It is worth noting the lack of inflammatory response in the PP* group (short PP exposure), supporting the idea that PP long-term feeding has deeper alterations in the fish gut and contributes to changes in the microbiota and an increase in fish mortality [35], in agreement with previous studies [34,58,59],

In so far as deficient diets could be considered a stress factor, long-term feeding could determine suppressive or depressive effects on the immune mechanisms [24,28,34,35]. The down-regulation of mRNA expression of some immune-related genes with the increased plant proteins in the diet has also been reported in other species [60]. On the other hand, lower gene expression values reported in the FM group might be related to a higher protection in the host from bacterial adhesion and growth. Nevertheless, results should be analyzed with caution, since a wide individual variation of inflammatory and immune genes expression has been reported in other species [61] and the level of expression before the ex vivo trial conditions the inflammatory and immune capacity registered after bacterial exposure [32].

Nevertheless, a longer stimulation with PP may lead to a degree of tolerance. This would explain why in the ex vivo challenges with pathogenic bacteria, stimulation in seabream samples from the PP* group was higher than in the PP group.

Finally, pathogenic bacteria have been selected for the challenge in the ex vivo assay due to their proven pathogenic activity in farmed gilthead seabream. *Photobacterium damselae* subsp. *Piscicida* (*P. damselae subsp. Piscicida*) is the causal agent of pasteurellosis (Romalde, 2002), *Pseudomonas anguilliseptica* (*P. anguilliseptica*) is related to “winter disease” [62], and *Vibrio alginolyticus* (*V. alginolyticus*) has been described as the causal agent of vibriosis [63], also associated with other *Vibrio* species in high mortality outbreaks [64]. This work showed that *V. alginolyticus* CECT 521 (ATCC 17749) had a very powerful inflammatory effect, displaying a very high induction of IL-1β. This is in line with the fact that the published genome of this strain displayed great toxigenic potential [65], with at least one gene encoding a pore forming RTX family toxin, among others (KEGG genomes, toxin search on assembly GCA_000354175.2).

## 4. Materials and Methods

### 4.1. Ethics Statement

The animal study protocol was reviewed and approved by the Universitat Politècnica de València Ethical Committee (code: P4-04-05-2017). All experiments were conducted in an accredited animal care facility (code: ES462500001091) in accordance with the guidelines and regulations set forth in Directive 2010/63/EU EEC and the Spanish Royal Decree 53/2013 on the protection of animals used for scientific purposes [66]

### 4.2. Fish, Rearing System Conditions, Diets, and Feeding Conditions

A total number of 240 juveniles of gilthead seabream (average weight 7.5 g and 60 days) were obtained from the fish farm, BERSOLAZ (Bersolaz Spain, S.L.U, Culmarex Group) located in Port de Sagunt (Valencia, Spain) and transported to the facilities at the Universitat Politècnica de València, where the growth trial was conducted after 15 days of adaptation to experimental conditions. Features of the system and water parameters set were described in previous growth trials carried out in these facilities [34,35]. Lighting conditions were determined by the natural photoperiod. Temperature, pH, oxygen, ammonia, nitrite, and nitrate concentrations were monitored throughout the period of the experiment. The fish were fed daily by hand to apparent satiation two times per day (9:00 and 17:00 h). The pellets were slowly distributed, allowing fish to eat, in a weekly regime of six days of feeding and one day of fasting.

Diets were prepared by a cooking extrusion process using a semi-industrial twin-screw extruder (CLEXTRAL BC-45, St. Etienne, France). A FM diet, in which most of the protein was provided by fish meal (59%), and a PP diet, in which all the fish meal was replaced by plant sources and synthetic amino acids were added to meet the minimum amino acid requirement for gilthead seabream juveniles [67]. Ingredients and proximate composition are shown in Table 2. Prior to diet formulation, dry matter, crude protein, crude lipid, ashes, and crude fiber (CF) of different sources and ingredients used were analyzed according to AOAC procedures [68]. All analyses were performed in triplicate. The amino acids of raw diets were also analyzed by reverse phase–high performance liquid chromatography [69]. Macronutrients and essential amino acid content were determined in the experimental diets, and they are shown in Table 2.

### 4.3. Bacterial Strains and Growth Conditions

Cultures of *P. anguilliseptica* CECT 901, *V. alginolyticus* CECT 521, and *P. damselae* subsp. *piscicida* CECT 7198 were obtained from the Colección Española de Cultivos Tipo (CECT, Valencia, Spain). The culture medium used for *P. anguilliseptica* was Tryptic Soy Broth (containing, 15.0 g tryptone, 5.0 g soy peptone, and 5.0 g NaCl, pH 7.3 per liter of water), and *V. alginolyticus* and *P. damselae* subsp. *piscicida* were grown on Marine Broth (containing, 5.0 g Bacto peptone, 1.0 g yeast extract, 0.10 g Fe(III) citrate, 19.45 g NaCl, 0.16 g Na_2_CO_3_, 3.24 g NaSO_4_, 1.80 g CaCl_2_, 8.80 g MgCl_2_, 0.55 g KCl, 0.08 g KBr, 34.00 mg SrCl_2_, 22.00 mg H_3_BO_3_, 4.00 mg Na-silicate, 2.40 mg NaF, 1.60 mg (NH_4_)NO_3_, and 8.00 mg Na_2_HPO_4_, pH 7.6 per liter of water). Bacteria were grown under agitation at 26 ℃ for 2 days (*P. anguilliseptica* and *P. damselae* subsp. *piscicida*) and at 30 ℃ for 1 day *(V. alginolyticus).* Then, 1.5 g/L of bacteriological agar were added to these media to prepare solid medium in petri dishes. For the bacterial challenge, the optical density (600 nm) of the bacterial cultures was determined and the bacterial cell number was estimated using the standard curves established for each strain. Then, bacterial cultures were centrifuged at 4,000 g for 20 min, washed once with PBS, and re-suspended in a CO_2_-independent cell culture medium (Gibco, ThermoFisher, Waltham, MA, USA) to a final concentration of 3 × 10^7^ ufc/mL in the case of *V. alginolyticus*, and 1 × 10^7^ ufc/mL of *P. damselae* subsp. *piscicida* and *P. anguilliseptica*.

### 4.4. Experimental Design

The aim of this work was to evaluate the impact of dietary fish protein substitution by plant protein on intestinal health status and its immune response capacity. For this purpose, fish were fed with FM or PP diets, and animals were sacrificed and processed at two critical growth phases of sea bream [16,17]: Phase I (90 days; from 12 to 68 g) and Phase II (305 days; up to 250 g). Each diet was assayed in tanks per triplicate. Figure 7 illustrates the experimental design. At each sampling time, intestine fragments were collected, part of them were used to determine basal expression of selected genetic markers, and other fragments were used in the culture explant procedure (ex vivo assay) to test the immune competence of the intestinal mucosa when challenged with different pathogenic bacteria.

#### 4.4.1. Phase I: Up to 68 g

Fish were fed with FM or PP diets, in tanks per triplicate (40 fish per tank), up to 90 days, being scattered 2 fish per group to be submitted to bacterial challenge (68 ± 37.8 g). For basal gene expression, two fragments from the FG and HG from each fish were placed in an Eppendorf tube containing 500 µL of RNA Later^®^ (Qiagen, Valencia, Spain) for subsequent total RNA (tRNA) extraction. Additionally, four fragments from FG and HG fragments were used for the ex vivo assay and exposed to the pathogens challenge (see below). Gene expression was determined in all samples to evaluate the inflammatory and immune status of the intestinal mucosa due to changes in the diet and the bacterial challenge.

#### 4.4.2. Phase II: Up to 250 g

Fish, ~30 fish per tank, were fed with the same diets, FM and PP, up to 305 days when the mean weight was 252 ± 70.1 g (Figure 1). In this second phase, in addition to the impact of long-term feeding with 100% of the PP diets, also a short-term exposure (15 days) of total fish meal substitution was evaluated. For this purpose, fish bred with the FM diet (*n* = ~15 fish per tank) were changed to a PP diet two weeks before the termination of the experiment (from day 290 to day 305) (PP* group). Three fish from the FM group and two from the PP and PP* were sacrificed to obtain FG and HG explants for ex vivo assays. As in the previous assay, for basal gene expression, two fragments from each fish were placed in RNA Later^®^ for subsequent total RNA (tRNA) extraction, and four pieces for ex vivo study.

##### Ex vivo Assays and Bacterial Challenge

Before tissue preparation, fish were sacrificed by immersion in benzocaine (60 ppm) during 15 min. Then, they were dissected and the intestine was obtained and separated in two sections (FG and HG). Each section was cut with a scalpel into small pieces (4 × 4 mm^2^), which were immediately placed in culture filter plates (15 mm diameter wells with 500 µm bottom-mesh, Netwell culture systems, Costar, Cambridge, MA, USA) with the epithelial surface facing up. Filters were placed into wells containing 1 mL of the different bacterial solutions (one of them was preserved without bacteria as control; ex vivo unchallenged group) in a CO_2_-independent cell culture medium (Gibco ThermoFisher; Waltham, MA, USA). A total of 100 µL of the corresponding bacterial solutions was finally added to the epithelial surface to ensure that samples were completely submerged. At the end of the incubation time, samples were carefully collected from the culture filter plates and stored in 100 mM Tris-HCl at 4 ℃ or RNA later at −80 ℃ for LDH activity evaluation or RNA isolation, respectively. Changes in pH of the explant culture medium due to different bacterial treatments were monitored. Explants of FG and HG from two fish per group were incubated during 4 and 6 h at 22 ℃ in independent CO_2_ atmosphere, depending on the experiment. The bacterial species used in the pathogen challenge were: *P. damselae* subsp., *P. anguilliseptica*, and *V. alginolyticus*. *P. anguilliseptica* was discarded in phase II, because bacterial concentration could not be determined due to aggregate formation. After explant assay, the samples were placed into RNA Later (Qiagen) for subsequent tRNA extraction. All conditions (fish/section/stimuli) were assayed in duplicate and gene expression was determined in all samples to evaluate the intestinal inflammatory and immune status based on experimental diet and bacterial challenge.

##### LDH Activity Assay

In order to determine tissue integrity, LDH activity was determined [70] in the tissue (U/mg protein) and explant culture medium (U/L) at different times of the incubation (0, 4, 6, and 24 h). LDH activity was analyzed measuring the nicotinamide adenine dinucleotide (NADH) absorbance at 340 nm using the commercial kit (BioSystems S.A., Barcelona, Spain). Tissue was weighed, homogenized in Tris-Hcl 100 mM while maintaining the tubes on ice, centrifuged at 12,000 rpm and 4 ℃ for 15 min, and supernatant was collected for LDH assessment. Total protein in tissue extracts was determined using Bradford [71].

##### Gene Expression Assay of Intestinal Inflammatory and Immune Markers

Based on the gene expression analysis used in previous studies of this species (*Sparus aurata*), to evaluate the intestinal inflammatory and immune status [35], tRNA was extracted from intestinal tissue samples using the phenol/chloroform method with Trizol Reagent (Invitrogen, Barcelona, Spain) and treated with DNAse I (Roche Diagnostics SL, Barcelona, Spain)) to remove DNases. Total RNA concentration, quality, and integrity were assessed using a NanoDrop 2000C Spectrophotometer (Fisher Scientific SL, Madrid, Spain). The integrity of 28S/18S was also determined by gel electrophoresis. An amount of 1 µg of total RNA was used for cDNA synthesis reaction using the qScript cDNA synthesis kit (Quanta BioScience, Beverly, MA, USA), according to the manufacturer’s instructions. An Applied Biosystems 2720 Thermal Cycler (ThermoFisher, Waltham MA, USA) was used with the following cycling conditions: 22 ℃ for 5 min, 42 ℃ for 30 min, and 85 ℃ for 5 min. cDNA samples were stored at −20 ℃ until gene expression was analyzed.

Four housekeeping candidate genes (Table 3) were tested to be used as reference genes and for assessing RNA integrity along the assay. The Cq of the four genes was determined in six pooled samples from Experiment 1 (two dietary groups: FM and PP; three times: 0, 4, and 6 h). Relative gene expression of six genes was determined in the FG and HG samples. The genetic markers monitored in this assay were three pro-inflammatory markers, *IL1-β*, *IL-6*, and *COX-2*, the main immunoglobulin, *IgM*, and the occludin gene, *Ocl*, with primers listed in Table 3.

All qPCR assays and expression analyses were performed using the Applied Biosystems 7500 Real-Time PCR with SYBR^®^ Green PCR Master Mix (ThermoFisher Scientific, Waltham, MA, USA). After an initial Taq activation of polymerase at 95 ℃ for 10 min, 42 cycles of PCR were performed with the following cycling conditions: 95 ℃ for 10 s and 60 ℃ for 30 s in all genes. In order to evaluate assay specificity, a melting curve analysis was directly performed after PCR cycles by slowly increasing the temperature (1 ℃/min) from 60 to 95 ℃, with continuous registration of changes in fluorescent emission intensity. The total volume for every PCR reaction was 20 μL, performed from a diluted (1:20) cDNA template (5 μL), forward and reverse primers (10 μM, 1 μL), SYBR^®^ Green PCR Master Mix (10 μL), ROX (2 μL, 10 nM), and nuclease-free water up to 20 μL. The analysis of the results was carried out using the 2^−∆∆Ct^ method. The target gene expression quantification was expressed relative to the expression of the selected reference gene. A cDNA pool from all the samples was included in each run and acted as a calibrator, and a non-template control for each primer pair, in which cDNA was replaced by water, was run on all plates. Reference and target genes in all samples were run in duplicate PCR reactions.

### 4.5. Statistics

Statistical data analysis was performed with Statgraphics© Centurion XVI software (Statistical Graphics Corp., Rockville, MO, USA).

LDH enzymatic activity in tissues and the supernatant was statistically analyzed by one-way analysis of variance (ANOVA) using the Newman–Keuls test to determine possible differences across the assay (0, 4, 6, and 24 h) in FG and HG.

The expression stability of reference genes was assessed using the BestKeeper program, based on the arithmetic means of the Cq values [72]. Lower deviation in the expression is related to better stability.

The evaluation of intestinal inflammatory and immune status was performed through the gene expression of the target genes in both in vivo and ex vivo conditions. The relative gene expression was statistically analyzed by ANOVA. Gene expression of cultured pieces was normalized with the expression of ex vivo unchallenged samples at 4 and 6 h. Multifactorial analysis was used to determine the significance (*p* < 0.05) of different factors considered (dietary treatment: FM/PP; intestinal section: FG/HG; bacterial stimuli: *P. damselae* subsp. *piscicida*/*P. anguilliseptica*/*V. alginolyticus*) at different times and to determine differences in normalized gene expression between dietary groups, sections, and bacterial stimuli, using the Newman–Keuls test. Data were expressed with the mean and the standard error of the normalized expression values, and differences were considered statistically significant when *p* < 0.05.

Additionally, with the aim of evaluating if the bacterial challenge is individually inducing the target genes or a combination of a set of genes, a correlation analysis was carried out and the Pearson product-moment coefficient was obtained for each pair of genes.

Finally, in order to confirm the assay reproducibility, the gene expression of the biological replicate samples was randomly assigned to different variables (*x* and *y*). Data consistency was evaluated for each gene by simple regression analysis using the model *y* = a*x*. Confidence intervals at 95% for a (a ± 1.96σ) were obtained for each gene to validate the hypothesis a = 1 (*y* = *x*).

## 5. Conclusions

PP appeared to alter the mucosal immune homeostasis. In the early stages of development, fish are very sensitive to plant diets, while adult fish seem to become tolerant to a constant PP diet, although maintaining a high threshold of inflammatory signals. The exposure to PP feed for a short term in adults led to a greater response to bacterial challenge. *V. alginolyticus* triggered the highest immune and inflammatory response. The successful evaluation of inflammatory and immune responses and pathogen challenge has been achieved by means of a new experimental system that implemented fish intestinal explants culture in gilthead seabream, a system that might be easily adapted to other teleost species. The use of such ex vivo methods constitutes an amenable technique that provides reliable results and helps in reducing the number of animals per assay.

## Figures and Tables

**Figure 1 ijms-21-07584-f001:**
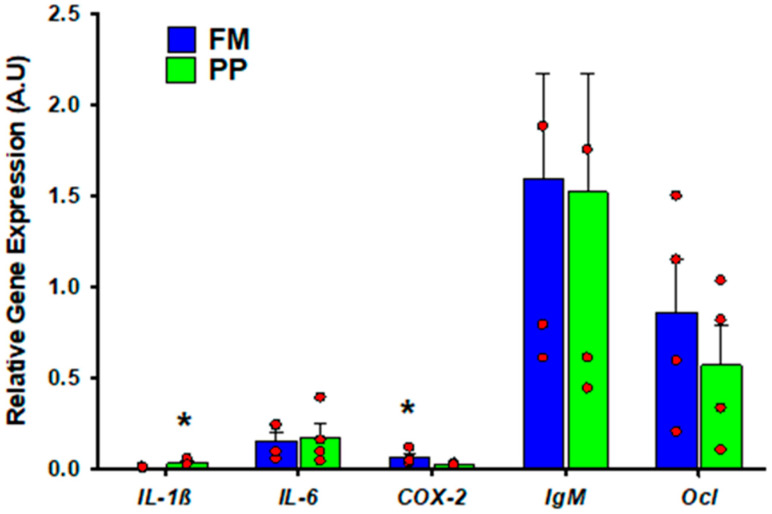
The intestinal basal expression of fish exposed to plant protein (PP) diet in Phase I (68 g). Relative gene expression (A.U.) of the different genes is expressed by the mean and standard error. Asterisks on the bars indicate significant differences (*p* < 0.05). Sample data in each experimental group are represented by red spots. Fish Meal (FM) diet. *Interleukin 1β: IL-1β; Interleukin 6:I L-6; Cyclooxygenase 2: COX-2; Immunoglobulin M: IgM; Occludin: Ocl*.

**Figure 2 ijms-21-07584-f002:**
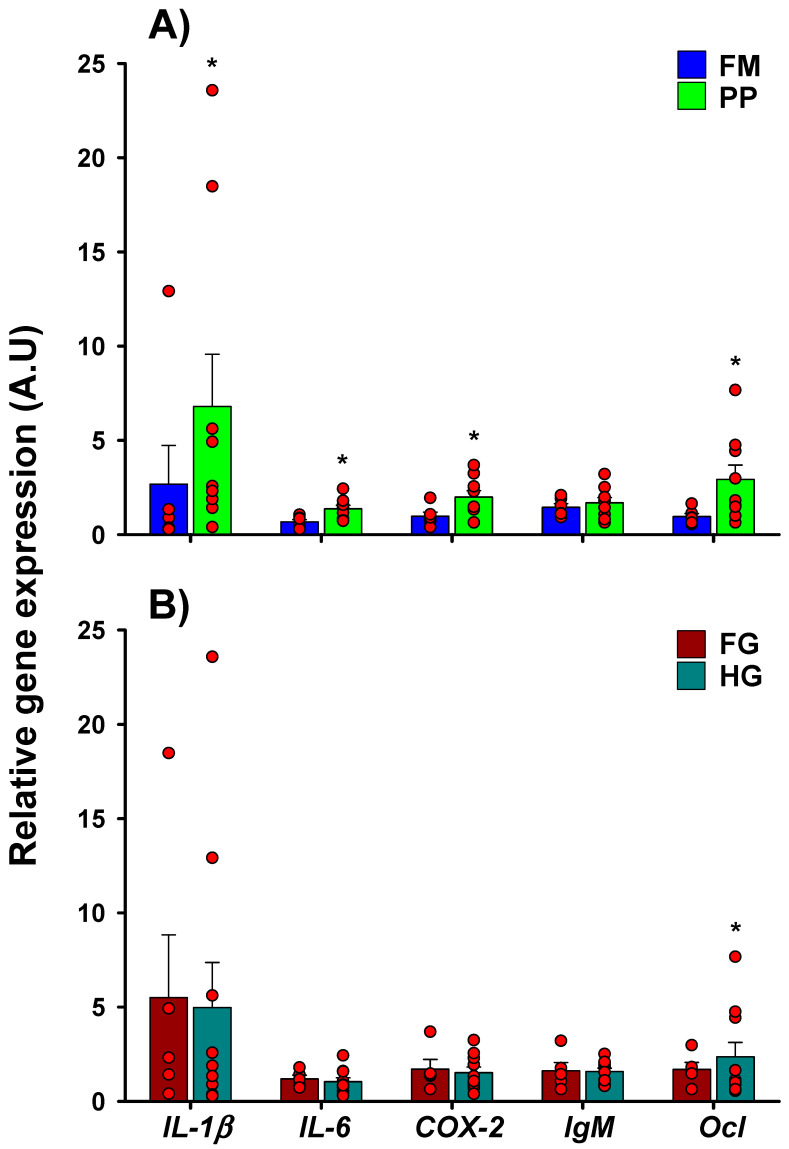
The effect of diet and intestinal section after 6 h of ex vivo bacterial exposition in Phase I (68 g). Relative gene expression (A.U.) of the different genes is expressed by the mean and standard error. Asterisks on the bars indicate significant differences between different conditions (diet/section) for each gene (*p* < 0.05) at 6 h of incubation. (**A**) Effect of dietary treatment (**B**) Effect of intestinal section. Sample data in each experimental group are represented by red spots. Foregut (FG); Hindgut (HD).

**Figure 3 ijms-21-07584-f003:**
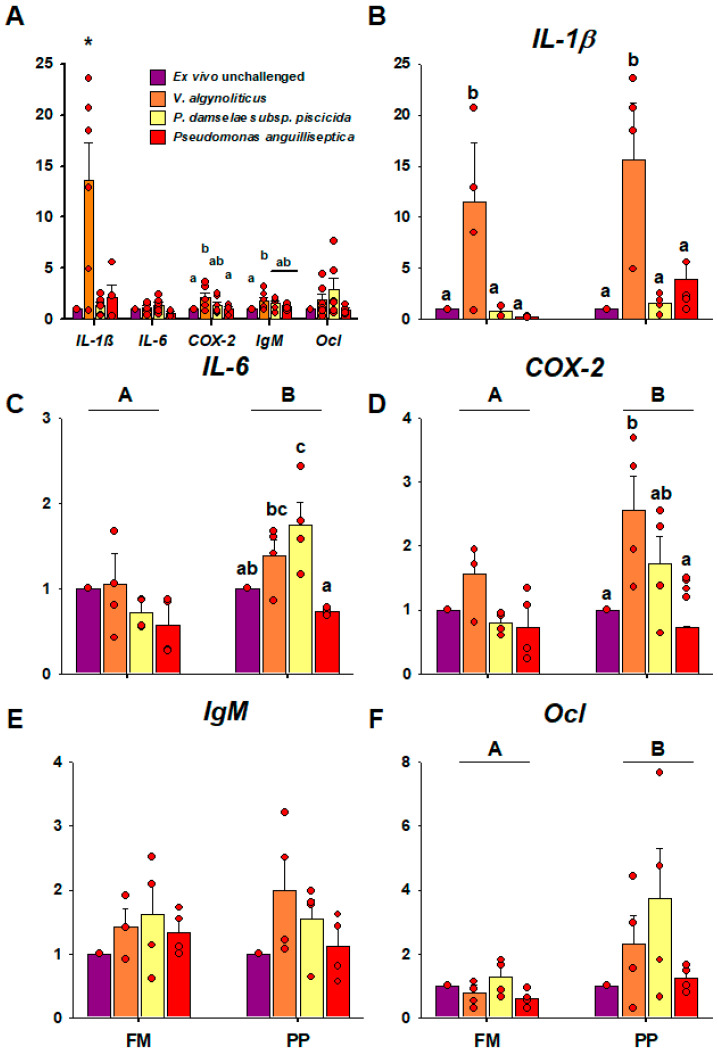
The effect of bacterial challenge based on the diet after 6 h of ex vivo bacterial exposition in Phase I (68 g). Relative gene expression (A.U.) of the different genes is expressed by the mean and standard error. Capital letters indicate differences between diets, meanwhile lowercase letters or asterisks indicate differences between bacteria for each gene (*p* < 0.05) after 6 h of incubation. (**A**) The effect of bacterial challenge independent of diet. Gene expression of (**B**) *IL-1β*, (**C**) *IL-6*, (**D**) *COX-2*, (**E**) *IgM*, and (**F**) *Ocl* based on the stimuli and diet. Sample data in each experimental group are represented by red spots. *Photobacterium damselae subsp. Piscicida* (*P. damselae subsp.*
*Piscicida*); *Pseudomonas anguilliseptica* (*P. anguilliseptica*); *Vibrio alginolyticus* (*V. alginolyticus*)

**Figure 4 ijms-21-07584-f004:**
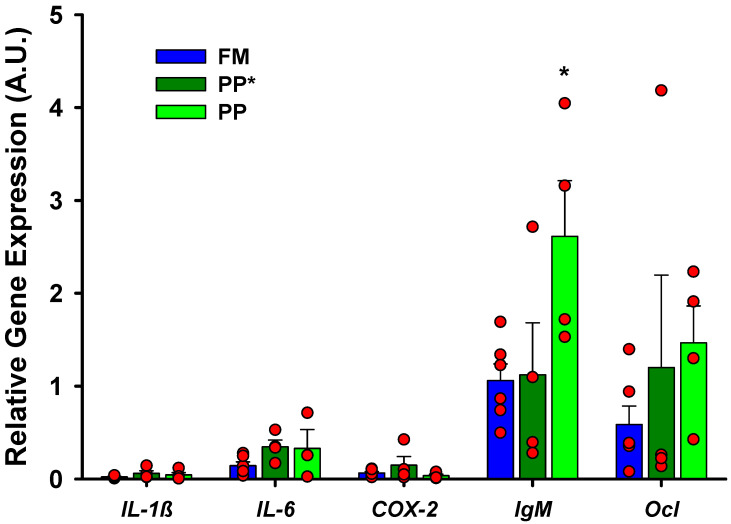
Intestinal basal gene expression of fish exposed to long (PP) or short term (PP*) diet in Phase II (250 g). The relative gene expression (A.U.) of the different genes is expressed by the mean and standard error. An asterisk on the bars indicates significant differences (*p* < 0.05). Sample data in each experimental group are represented by red spots.

**Figure 5 ijms-21-07584-f005:**
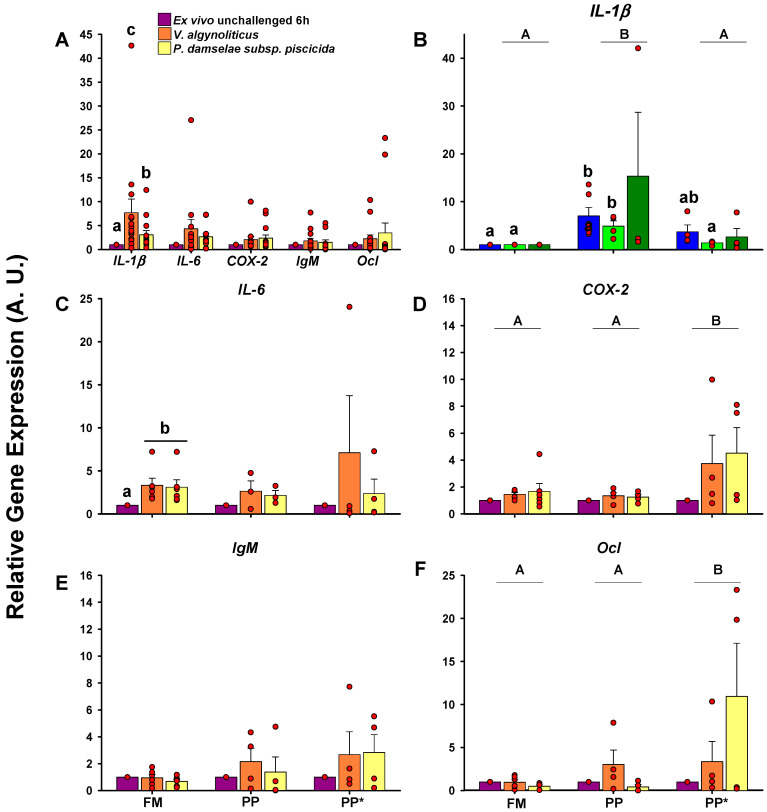
The effect of bacterial challenge based on diet after ex vivo bacterial exposition in Phase II (250 g). Relative gene expression (A.U.) of the different genes is expressed by the mean and standard error. Capital letters indicate differences between diets, meanwhile lowercase letters indicate differences between bacteria for each gene (*p* < 0.05) after 6 h of incubation. (**A**) The effect of bacterial challenge independent of diet. Gene expression of (**B**) *IL-1β*, (**C**) *IL-6,* (**D**) *COX-2*, (**E**) *IgM*, and (**F**) *Ocl* based on the stimuli and diet. Sample data in each experimental group are represented by red spots.

**Figure 6 ijms-21-07584-f006:**
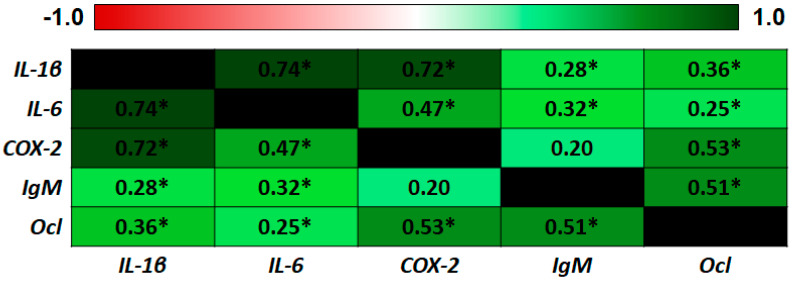
Correlation analysis of gene expression determined in samples after the ex vivo assay. Pearson product-moment coefficients between each pair of genes. Significant correlations are indicated with an *.

**Figure 7 ijms-21-07584-f007:**
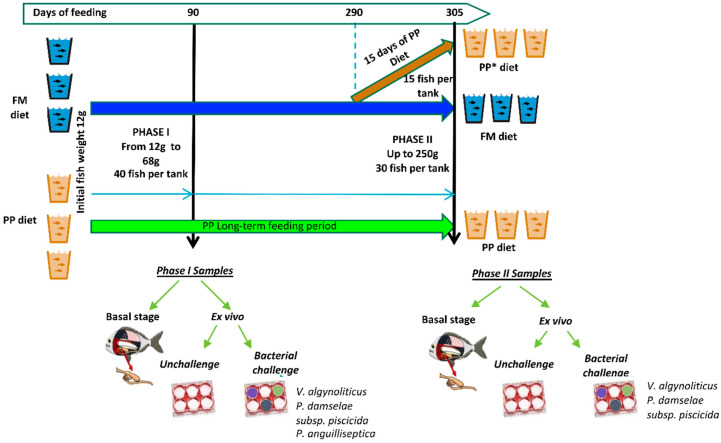
Summary of the experimental design. The impact of dietary fish protein substitution by plant protein in fish immune response to the ex vivo bacterial challenge was evaluated at two on-growing phases: 12–68 g (90 days) and up to 250 g (305 days). Additionally, a group was included at 305 days to estimate the effect of short-term fish meal substitution (15 days; PP*).

**Table 1 ijms-21-07584-t001:** Effect of different factors on normalized gene expression values in Phase II.

	*IL-1β*	*IL-6*	*COX-2*	*IgM*	*Ocl*
Diet	0.533	0.601	0.025 *	0.120	0.044 *
Section	0.157	0.138	0.168	0.864	0.486
Stimuli	0.003 *	0.036 *	0.021 *	0.218	0.163

*p*-values obtained for each factor in the multifactorial analysis. Significant values are indicated by *. Interleukin 1β: IL-1β; Interleukin 6:I L-6; Cyclooxygenase 2: COX-2; Immunoglobulin M: IgM; Occludin: Ocl.

**Table 2 ijms-21-07584-t002:** Ingredients (g kg^−1^ as fed) and proximate composition (% dry weight) of the experimental diets *.

	Diet	
Ingredients (g kg^−1^)	FM	PP
Fish meal	589	
Wheat meal	260	
Wheat gluten		295
Broad bean meal		41
Soybean meal		182
Pea meal		41
Sunflower meal		158
Krill meal		
Squid meal		
Fish oil	38.1	90
Soybean oil	92.9	90
Soy Lecithin	10	10
Vitamin-mineral mix ^1^	10	10
Calcium phosphate		38
Arginine		5
Lysine		10
Methionine		7
Taurine		20
Threonine		3
Proximate composition (% dry weight)		
Dry matter	88.1	93.9
Ashes	10.1	7.4
Crude lipid	18.5	19.8
Crude fiber	0.8	4.3
Crude protein	44.2	45.0
Essential amino acids (g 100 g^-1^)		
Arginine	3.39	3.30
Histidine	1.00	0.82
Isoleucine	1.47	1.17
Leucine	3.24	2.98
Lysine	3.68	2.26
Methionine	1.16	1.06
Phenylalanine	1.80	1.87
Threonine	1.98	1.44
Valine	2.01	1.47

* Fish meal (FM) diet: Diet formulated with fish meal as protein source; Plant protein (PP) diet: Diet in which fish meal was totally substituted with plant protein mixture. ^1^ Vitamins and mineral mixture (values are g kg^−1^): Premix, 25; Hill, 10; DL-a- tocopherol, 5; ascorbic acid, 5; (PO_4_)_2_Ca_3_, 5. Premix composition (values are IU kg^−1^): Retinol acetate, 1,000,000; calciferol, 500; DL-a-tocopherol, 10; menadione sodium bisulfite, 0.8; hydrochlorhydrate thiamine, 2.3; riboflavin, 2.3; pyridoxine hydrochloride, 15; cyanocobalamin, 25; nicotinamide, 15; pantothenic acid, 6; folic acid, 0.65; biotin, 0.07; ascorbic acid, 75; inositol, 15; betaine, 100; polypeptides, 12.

**Table 3 ijms-21-07584-t003:** Primer sequences of candidate genes (reference and target genes) in the RT-qPCR assay.

Gene ^1^	GeneBank ID	Primer Forward (5′→3′)	Primer Reverse (5′→3′)	Length (pb)	Reference
	REFERENCE GENES				
*EF-1α*	AF184170	CTGTCAAGGAAATCCGTCGT	TGACCTGAGCGTTGAAGTTG	87	[35,36]
*GAPDH*	DQ641630	CCAACGTGTCAGTGGTTGAC	AGCCTTGACGACCTTCTTGA	80	[37]
*Rps18*	AM490061	AGGGTGTTGGCAGACGTTAC	CGCTCAACCTCCTCATCAGT	97	[37]
*β-Act*	X89920	TCTGTCTGGATCGGAGGCTC	AAGCATTTGCGGTGGACG	113	[38]
	TARGET GENES				
*IL-1β*	AJ277166	GCGACCTACCTGCCACCTACACC	TCGTCCACCGCCTCCAGATGC	131	[37]
*IL-6*	AM749958	AGGCAGGAGTTTGAAGCTGA	ATGCTGAAGTTGGTGGAAGG	101	[35]
*COX-2*	AM296029	GAGTACTGGAAGCCGAGCAC	GATATCACTGCCGCCTGAGT	192	[55]
*IgM*	JQ811851	TCAGCGTCCTTCAGTGTTTATGATGCC	CAGCGTCGTCGTCAACAAGCCAAGC	131	[39]
*Ocl*	JK692876	GTGCGCTCAGTACCAGCAG	TGAGGCTCCACCACACAGTA	81	[35,36]

^1^ Elongation Factor 1α: EF-1α; Glyceraldehyde 3-phosphate dehydrogenase: GAPDH; Ribosomal Protein S18: Rps18; β-Actin: β-Act; Interleukin 1β: IL-1β; Interleukin 6:I L-6; Cyclooxygenase 2: COX-2; Immunoglobulin M: IgM; Occludin: Ocl.

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
