# Peer review of "Intestinal Explant Cultures from Gilthead Seabream (*Sparus aurata, L.)* Allowed the Determination of Mucosal Sensitivity to Bacterial Pathogens and the Impact of a Plant Protein Diet"

_ijms, 2020, doi:10.3390/ijms21207584_

Round 1

Reviewer 1 Report

The manuscript by Penaranda DS et al. reports the impact of substituting a fish meal diet by a plant protein diet on the expression of genes involved in the immune- and inflammatory response in the gut of the gilthead seabream, a commercially farmed fish. More specifically, it employs intestinal explant cultures to study the gene expression in the gut of immune/inflammatory markers under two different diets and in response to various bacterial challenges. The main finding is that a plant diet triggers a higher inflammatory response to bacterial challenge with a positive correlation between the proinflammatory marker genes IL1-beta, IL-6 and COX-2.

General:

The research group has previously published several papers on this topic (duly cited in the text), and the conclusions of the present work regarding dietary effects on the inflammatory marker genes named above are largely confirmatory of earlier work. The novelty here is the data obtained using ex-vivo intestinal explant culture with exposure to bacterial challenges. This might be potentially interesting, but reporting changes in relative gene expression levels of selected markers remains solely descriptive if no functional implications are presented as well. For instance, the work would be much strengthened if changes at the protein level (i.e. western blots) were included in the study. In addition, and as explained below, I am not fully convinced that explant culture is a reliable model for this kind of study.    

The English language needs thorough editing throughout the text because in many instances the precise meaning of sentences is obscure to the reader.

Specific points:

There is a discrepancy between in the numbering order of the figures (figs. 1/7) in the manuscript and in the supplementary files.

Questions to the layout of figures: In Fig. 2A, the labeling of the genes apparently is missing. Also, this figure shows only five genes, not six as in Fig. 1 and Fig. 2B. Fig. 1 vs. Fig. 2B: Finally, the order of the genes iMuc and IgM is switched around.  

Fig. 4 and line 138: "No gene expression differences were observed…., except for IgM…": Yet, the figure shows that iMuc was drastically reduced in both PP groups (although no asterisk indicates significance). This needs to be explained. Furthermore, in line 151, it is stated that iMuc data had a high variability between duplicates and therefore "was not considered in further analysis." If this is the case, why include iMuc in the study in the first place?

Fig. 3 and Fig. 5: From Figs. S3 and S7 it is evident that the relative expression of the proinflammatory marker genes increases steeply by ex-vivo culture itself and with a high statistical uncertainty. In my view, it questions the validity of the reported effects of bacterial challenges measured by use of the explant culture when this model system by itself profoundly affects the parameters under study. The multifactorial statistics used by the authors to assess the significance of the measured effects may well be appropriate, but I am nevertheless less than fully convinced about the overall reliability of these data. Furthermore, why are the ex-vivo data obtained from the Phase I (Fig. 3) and Phase II (Fig. 5) experiments not presented in the same way?

Line 690: Ref. 73 is incorrectly presented.

Author Response

Dear Reviewer,

Enclosed is the revised version of our manuscript entitled “Intestinal explant cultures from gilthead seabream (Sparus aurata, L.) allowed the determination of mucosal sensitivity to bacterial pathogens and the impact of a plant protein diet” (Manuscript ID: ijms-917349).

We have responded to all of the comments and our responses are laid out below. On behalf of all co-authors I appreciate all the comments that were given and feel that through the critical revision process, we have strengthened the integrity of the study and provided a stronger manuscript.

Comments and Suggestions for Authors

The manuscript by Peñaranda DS et al. reports the impact of substituting a fish meal diet by a plant protein diet on the expression of genes involved in the immune- and inflammatory response in the gut of the gilthead seabream, a commercially farmed fish. More specifically, it employs intestinal explant cultures to study the gene expression in the gut of immune/inflammatory markers under two different diets and in response to various bacterial challenges. The main finding is that a plant diet triggers a higher inflammatory response to bacterial challenge with a positive correlation between the proinflammatory marker genes IL1-beta, IL-6 and COX-2.

General:

  • The research group has previously published several papers on this topic (duly cited in the text), and the conclusions of the present work regarding dietary effects on the inflammatory marker genes named above are largely confirmatory of earlier work. The novelty here is the data obtained using ex-vivo intestinal explant culture with exposure to bacterial challenges. This might be potentially interesting, but reporting changes in relative gene expression levels of selected markers remains solely descriptive if no functional implications are presented as well. For instance, the work would be much strengthened if changes at the protein level (i.e. western blots) were included in the study. In addition, and as explained below, I am not fully convinced that explant culture is a reliable model for this kind of study.

Answer: We understand that protein analysis would add some valuable information; however, under our point of view, it would not provide an essential information that we can not obtain with the gene expression about the functional implications of these genes after bacterial challenge. On the other hand, the reliability is based on the statistical evidences. These evidences are presented in the manuscript: tissue integrity, effect of the methodology, time of incubation and data consistency. Of course, further and more extended experiments could have been done, but we consider that the provided data are enough to validate the explant culture developed.

  • The English language needs thorough editing throughout the text because in many instances the precise meaning of sentences is obscure to the reader.

Answer: The English of the manuscript has been reviewed by a professional editor. Please, see the certificate that the manuscript was edited and revised for proper English language, syntax, spelling, punctuation and style by David Harry Rhead (B.A. Modern Languages and International Studies)..

Specific points:

  • There is a discrepancy between in the numbering order of the figures (figs. 1/7) in the manuscript and in the supplementary files.

Answer: It has been corrected.

  • Questions to the layout of figures: In Fig. 2A, the labeling of the genes apparently is missing. Also, this figure shows only five genes, not six as in Fig. 1 and Fig. 2B. Fig. 1 vs. Fig. 2B: Finally, the order of the genes iMuc and IgM is switched around.

Answer: This was certainly our mistake, the missing gene in Figure 2A has been included. The order of the genes in Figure 1 and 2 is now the same. On the order hand, as the genes in both Figures: 2A and 2B are the same, we have decided to include the labelling of the genes only in the Figure 2B, with the purpose of keeping the figure as simple as possible.

  • 4 and line 138: "No gene expression differences were observed…., except for IgM…": Yet, the figure shows that iMuc was drastically reduced in both PP groups (although no asterisk indicates significance). This needs to be explained.

Answer: As the size of sampling group changes depending of the variables that has been taking into account, the data distribution has been included in the figures in order to help to the reader to understand the statistical differences come from. iMuc expression has been removed from the study (see below).

  • Furthermore, in line 151, it is stated that iMuc data had a high variability between duplicates and therefore "was not considered in further analysis." If this is the case, why include iMuc in the study in the first place?

Answer: Following the reviewer suggestion, we have removed the iMuc from the analysis.

  • 3 and Fig. 5: From Figs. S3 and S7 it is evident that the relative expression of the proinflammatory marker genes increases steeply by ex-vivo culture itself and with a high statistical uncertainty. In my view, it questions the validity of the reported effects of bacterial challenges measured by use of the explant culture when this model system by itself profoundly affects the parameters under study.

Answer: Certainly, during the ex vivo culture procedure, expression of proinflammatory marker genes increased significantly with the time of incubation, a phenomenon already described and most probably due to tissue injury (Harms and Lewbart 2000; Di Paolo and Shayakhmetov 2016). This induction in the controls is inherent to the system, and it must be described and taken into account, but it is relatively small. Of note, these values of inflammatory markers in Figures S3 and S5 correspond to the control of graphs in Figures 3 and 5, where they become the reference value, that is 1.00.

See lines 86-88: On the other hand, if the effect of ex vivo procedure is evaluated, significant differences were observed in most of the genetic markers analysed (Figure S3). Therefore, in the following assays, gene expression was normalised based on the ex vivo unchallenged samples.

See lines 151-154: The expression of pro-inflammatory genes (IL-6 and COX-2 genes) increased after 6 h of incubation in the ex vivo unchallenged group respect to the basal values, confirming the results of previous assay (Figure S7). Hence, expression results in samples incubated with the different bacteria were normalised with the expression of the control samples, for each experimental factor.

After the normalization of ex vivo unchallenged gene expression, significant data induced by bacterial challenge were registered in the range of 2 - 10 fold respect control group (normalized ex vivo unchallenged). Based on these results, we considered the explant culture developed as a reliable technique.

  • The multifactorial statistics used by the authors to assess the significance of the measured effects may well be appropriate, but I am nevertheless less than fully convinced about the overall reliability of these data. Furthermore, why are the ex-vivo data obtained from the Phase I (Fig. 3) and Phase II (Fig. 5) experiments not presented in the same way?

Answer: As mentioned above, reliability of the system relies on the difference between the expression markers: there is much greater induction in the samples than the induction introduced by the procedure. Fig. 3 (Phase I) and Fig. 5 (Phase II) have been modified and now are presented in the same way following the reviewer suggestion.

  • Line 690: Ref. 73 is incorrectly presented.

Answer: Thank you for your comment. The Ref 73 has been modified:  “Bradford, M.M. A rapid and sensitive method for the quantitation of microgram quantities of protein utilizing the principle of protein-dye binding. Anal. Biochem. 1976, 72, 248–254, doi:10.1016/0003-2697(76)90527-3.”

Cited references

Harms, C. A., and G. A. Lewbart. 2000. “Surgery in Fish.” The veterinary clinics of North America. Exotic animal practice 3(3): 759–74.

Di Paolo, Nelson C., and Dmitry M. Shayakhmetov. 2016. “Interleukin 1α and the Inflammatory Process.” Nature Immunology 17(8): 906–13.

Reviewer 2 Report

Thank you for the opportunity to review your research on the innate immune gut response against an ex vivo bacterial challenge after a total substitution of fish meal by plant protein in two seabream growth stages. The author conducted a long period of study covering two important phases of the seabream life cycle. Just wondering why the author decided to substitute total fishmeal with plant protein. It is already well-established that beyond certain levels of plant protein cause a deleterious effect on carnivorous fish which require a high amount of protein. Just wondering why the author did not consider including graded levels of plant protein in the diet of seabream that could provide a better understanding on effect of gradual increment of plant protein.

Yes, younger fish will be very sensitive to the total substitution of plant protein since plant protein contains many ANFs and also sometimes lacking essential amino acids and even some essential minerals though the author has added some mineral premixes in both of the diets. 

In reviewing your current submission I came across another published article by your group investigating very similar research questions with plant-based proteins (i.e. Long-term feeding with high plant protein-based diets in gilthead seabream (Sparusaurata, L.) leads to changes in the inflammatory and immune-related gene expression at intestinal level; POLS ONE April 30 2020). In this published paper, authors have already proved the negative consequence of adding plant-based proteins on seabream in terms of pro-inflammatory mediators (il1β,il6and cox2)and immune-related molecules (igm and alp) and so on.

Yes, intestinal explant culture seems to me a new method but author could do some histological analysis in the liver, spleen, kidney, intestine, and muscle that could give more insight along with explant culture and inflammatory response. This, together with a small number of analyses such as explant culture and RT- qPCR analysis form an unsatisfactory design.

I have taken the liberty of commenting directly on the submitted manuscript. I hope some of my suggestions and queries will help to improve your MS. Thank you.

Author Response

Dear Reviewer,

Enclosed is the revised version of our manuscript entitled “Intestinal explant cultures from gilthead seabream (Sparus aurata, L.) allowed determining the mucosal sensitivity to bacterial pathogens and the impact of a plant protein diet” (Manuscript ID: ijms-917349).

We have responded to all of the comments and our responses are laid out below. On behalf of all co-authors I appreciate all the comments that were given and feel that through the critical revision process, we have strengthened the integrity of the study and provided a stronger manuscript.

Comments and Suggestions for Authors

  • Thank you for the opportunity to review your research on the innate immune gut response against an ex vivo bacterial challenge after a total substitution of fish meal by plant protein in two seabream growth stages. The author conducted a long period of study covering two important phases of the seabream life cycle. Just wondering why the author decided to substitute total fishmeal with plant protein. It is already well-established that beyond certain levels of plant protein cause a deleterious effect on carnivorous fish which require a high amount of protein. Just wondering why the author did not consider including graded levels of plant protein in the diet of seabream that could provide a better understanding on effect of gradual increment of plant protein.

Answer: Our group studied in the past different levels of FM substitution in this species (Baeza-Ariño et al., 2016; Monge-Ortíz et al., 2016; Estruch et al., 2015, 2018a, 2018b), reporting that the fish fed with a 100% of PP diet showed the most altered intestinal health. Nevertheless, in this work we tried to set up a new perspective to understand the mechanisms by which the plant protein is affecting the intestinal health of this species. As previous data suggested that the effect of PP may escape from linearity, we considered that in this pioneering set of assays, maximizing differences was a priority. Further, there was a very important issue concerning the length of time and age that fish are exposed to plant based diet that could be essential in aquaculture. So, for reasons related to technical limitations of the procedures we had to prioritize at this point assaying short vs. long time of exposure in different ages, above PP gradients. We definitely agree that, for practical reasons, it would be essential to determine in future experiments the threshold at which plant protein begins to damage the immune system.

  • Yes, younger fish will be very sensitive to the total substitution of plant protein since plant protein contains many ANFs and also sometimes lacking essential amino acids and even some essential minerals though the author has added some mineral premixes in both of the diets.

Answer: The reviewer comment targets very important issues in all attempts to replace fish meal with plant protein, from the sensitivity of younger fish to the likely presence of ANFs and nutrient deficiencies. In this assay, plant diet was supplemented with synthetic AA to reach the optimal AA requirements of seabream juveniles (Peres and Oliva-Teles, 2009). Unfortunately, the optimal minerals and vitamins requirements have not yet been fully defined in fish (except some of them such as P, Ca…). Despite these constraints, in previous studies, in diets with different levels of FM substitution was added an equal quantity of vitamins and mineral without a decrement in terms of growth and survival (Benedito-Palos et al., 2016; Yaghoubi et al., 2016; Kriton et al., 2018; Mirghaed et al., 2019; Choi et al., 2020). Therefore, in the current manuscript, we followed the same procedure.

  • In reviewing your current submission I came across another published article by your group investigating very similar research questions with plant-based proteins (i.e. Long-term feeding with high plant protein-based diets in gilthead seabream (Sparus aurata, L.) leads to changes in the inflammatory and immune-related gene expression at intestinal level; POLS ONE April 30 2020). In this published paper, authors have already proved the negative consequence of adding plant-based proteins on seabream in terms of pro-inflammatory mediators (il1β,il6and cox2)and immune-related molecules (igm and alp) and so on.

Answer: Our previous experience has been essential to design this work, since these diets showed alterations in the intestinal health through the gene expression. Nevertheless, our aim in the current study was to focus on the processes that lead to the susceptibility of fish groups on plant protein diet to bacterial challenge, through the use of an ex vivo procedure, after a long and short feeding period with 100% vegetable protein diet. We used the same set of marker genes than in other trials to favor the comparisons of the results from the ex vivo assays with other feeding assays with plant protein diets.

  • Yes, intestinal explant culture seems to me a new method but author could do some histological analysis in the liver, spleen, kidney, intestine, and muscle that could give more insight along with explant culture and inflammatory response. This, together with a small number of analyses such as explant culture and RT- qPCR analysis form an unsatisfactory design. 

Answer: In previous in vivo studies using these diets, histological studies showed that the PP group only had thinner villi and lamina propria, but other differences were not found in the rest of parameters (serous, muscular and submucous layer or villi length and goblet cells; (Estruch et al., 2018b). Due to the lack of relevant differences, we decided not to include a histological analysis in the current manuscript. Plant protein induced higher mortality without differences in growth rate (Estruch et al. 2015; Estruch et al. 2018b).The purpose of this work was to test if ex vivo assays would help monitoring alterations in the immune system more clearly than the histological preparations did in the past. Additionally, we included an experimental group fed with PP diet for short period, demonstrating that even a few weeks of feeding without FM protein source has an effect on intestinal immune status.

On the other hand, the reason to perform this analysis in ex vivo conditions based on the limitations of in vivo assays. In vivo challenges with bacteria require specific facilities, increasing operative costs and a higher number of fish. Therefore, in the present manuscript, an effective ex vivo system has been developed and it also proved to be a useful tool to evaluate effect of different pathogen strains on the intestine that provided reliable information about the interactions between the bacteria and the host. Through the ex vivo method developed here it will be possible to maintain specific and controlled assay conditions and, in addition, to reduce the number of experiments and fish used, following the EU recommendations.

Cited references

Baeza-Ariño, R. et al. (2016) ‘Study of liver and gut alterations in sea bream, Sparus aurata L., fed a mixture of vegetable protein concentrates’, Aquaculture Research, 47(2), pp. 460–471. doi: 10.1111/are.12507.

Benedito-Palos, L. et al. (2016) ‘Lasting effects of butyrate and low FM/FO diets on growth performance, blood haematology/biochemistry and molecular growth-related markers in gilthead sea bream (Sparus aurata)’, Aquaculture. Elsevier B.V., 454, pp. 8–18. doi: 10.1016/j.aquaculture.2015.12.008.

Choi, D. G. et al. (2020) ‘Replacement of fish meal with two fermented soybean meals in diets for rainbow trout ( Oncorhynchus mykiss )’, Aquaculture Nutrition. Blackwell Publishing Ltd, 26(1), pp. 37–46. doi: 10.1111/anu.12965.

Estruch, G. et al. (2015) ‘Impact of fishmeal replacement in diets for gilthead sea bream (Sparus aurata) on the gastrointestinal microbiota determined by pyrosequencing the 16S rRNA gene’, PLoS ONE, 10(8). doi: 10.1371/journal.pone.0136389.

Estruch, G. et al. (2018a) ‘Inclusion of alternative marine by-products in aquafeeds with different levels of plant-based sources for on-growing gilthead sea bream (Sparus aurata, L.): effects on digestibility, amino acid retention, ammonia excretion and enzyme activity’, Archives of Animal Nutrition. Taylor & Francis, 72(4), pp. 1–19. doi: 10.1080/1745039X.2018.1472408.

Estruch, G. et al. (2018b) ‘Long-term feeding with high plant protein based diets in gilthead seabream (Sparus aurata, L.) leads to changes in the inflammatory and immune related gene expression at intestinal level’, BMC Veterinary Research. BioMed Central, 14(1), p. 302. doi: 10.1186/s12917-018-1626-6.

Kriton, G. et al. (2018) ‘ Impact of Diets Containing Plant Raw Materials as Fish Meal and Fish Oil Replacement on Rainbow Trout (Oncorhynchus mykiss) , Gilthead Sea Bream (Sparus aurata) , and Common Carp (Cyprinus carpio) Freshness ’, Journal of Food Quality. Hindawi Limited, 2018, pp. 1–14. doi: 10.1155/2018/1717465.

Mirghaed, A. T. et al. (2019) ‘Dietary sodium butyrate (Butirex® C4) supplementation modulates intestinal transcriptomic responses and augments disease resistance of rainbow trout (Oncorhynchus mykiss)’, Fish and Shellfish Immunology. Academic Press, 92, pp. 621–628. doi: 10.1016/j.fsi.2019.06.046.

Monge-Ortíz, R. et al. (2016) ‘Potential use of high levels of vegetal proteins in diets for market-sized gilthead sea bream (Sparus aurata)’, Archives of Animal Nutrition, 70(2), pp. 155–172. doi: 10.1080/1745039X.2016.1141743.

Peres, H. and Oliva-Teles, A. (2009) ‘The optimum dietary essential amino acid profile for gilthead seabream (Sparus aurata) juveniles’, Aquaculture, 296(1–2), pp. 81–86. doi: 10.1016/j.aquaculture.2009.04.046.

Yaghoubi, M. et al. (2016) ‘Dietary replacement of fish meal by soy products (soybean meal and isolated soy protein) in silvery-black porgy juveniles (Sparidentex hasta)’, Aquaculture. Elsevier B.V., 464, pp. 50–59. doi: 10.1016/j.aquaculture.2016.06.002.

Round 2

Reviewer 1 Report

No comments